# The Functional Neuroimaging of Autobiographical Memory for Happy Events: A Coordinate-Based Meta-Analysis

**DOI:** 10.3390/healthcare12070711

**Published:** 2024-03-24

**Authors:** Giulia Testa, Igor Sotgiu, Maria Luisa Rusconi, Franco Cauda, Tommaso Costa

**Affiliations:** 1Instituto de Transferencia e Investigación, Universidad Internacional de La Rioja, 26004 La Rioja, Spain; 2Department of Human and Social Sciences, University of Bergamo, 24129 Bergamo, Italy; igor.sotgiu@unibg.it (I.S.); marialuisa.rusconi@unibg.it (M.L.R.); 3Department of Psychology, University of Turin, 10124 Turin, Italy; franco.cauda@unito.it (F.C.); tommaso.costa@unito.it (T.C.); 4GCS-fMRI Research Group, Koelliker Hospital, 10134 Turin, Italy

**Keywords:** activation likelihood estimation meta-analysis, autobiographical memory, functional magnetic resonance imaging, autobiographical recall, happy life events

## Abstract

Neuroimaging studies using autobiographical recall methods investigated the neural correlates of happy autobiographical memories (AMs). The scope of the present activation likelihood estimation (ALE) meta-analysis was to quantitatively analyze neuroimaging studies of happy AMs conducted with autobiographical recall paradigms. A total of 17 studies (12 fMRI; 5 PET) on healthy individuals were included in this meta-analysis. During recall of happy life events, consistent activation foci were found in the frontal gyrus, the cingulate cortex, the basal ganglia, the parahippocampus/hippocampus, the hypothalamus, and the thalamus. The result of this quantitative coordinate-based ALE meta-analysis provides an objective view of brain responses associated with AM recollection of happy events, thus identifying brain areas consistently activated across studies. This extended brain network included frontal and limbic regions involved in remembering emotionally relevant positive events. The frontal gyrus and the cingulate cortex may be responsible for cognitive appraisal processes during recollection of happy AMs, while the subthalamic nucleus and globus pallidus may be involved in pleasure reactions associated with recollection of happy life events. These findings shed light on the neural network involved in recalling positive AMs in healthy individuals, opening further avenues for future research in clinical populations with mood disorders.

## 1. Introduction

Autobiographical memory (AM) refers to the ability to remember events and lifetime periods from one’s past, which is crucial for a sense of identity, self-continuity, and mental time traveling [1,2,3,4]. AM is considered a subsystem of episodic memory, more generally defined as the conscious recollection of experienced events and meaningful reconstruction of one’s own past [5]. According to Rubin [6,7,8], retrieval of AMs results from the interaction between multiple neural and cognitive systems; specifically, this process includes the recollection of personal life events (i.e., the ability to retrieve, re-experience, and relive a past event), self-referential processing, mental imagery, narrative reasoning, language, and emotion. 

AMs are generally characterized by emotional content compared to other types of episodic or semantic memories [1,9,10]. AMs with positive emotional valence are generally marked under the broad category of positive AMs. However, the expressions positive AMs and happy AMs are often used interchangeably in the literature. In the following, we will refer to the concept of happy AMs. Indeed, the majority of the studies included in the current meta-analysis instructed participants to recall happy AMs instead of positive AMs. Noteworthy, happy life events do not constitute a homogenous category: for example, a distinction between hedonic and eudaimonic happy events (hedonic events include life occurrences in which people pursue extrinsically motivated activities to experience enjoyment and pleasure, either sensory or psychological, whereas eudaimonic happy events are life occurrences in which people engage in intrinsically meaningful activities that enable the person to cultivate his or her skills and to develop his or her best potentials [11,12]) has been proposed in recent years by Positive Psychology [13,14,15]. Happy personal events occur frequently and occupy a central position in the life stories of individuals [11,12,16,17]. Empirical research on happy AMs may be particularly relevant since remembering these types of events may induce the re-experience of positive emotions, potentially contributing to psychological well-being and quality of life. 

Functional neuroimaging studies of AM increased over the last decade, allowing an enlarged understanding of brain processes and neural underpinning of memory for personal experiences. A distributed network encompassing different subsystems of AMs (e.g., recollection, self-referential processing, emotional component) has been identified. This network operates thanks to the contribution of the following neural areas: the medial temporal lobe and the hippocampus for AMs retrieval [18]; the lateral prefrontal cortex (lPFC) for memory search and controlled processes [19,20]; the medial prefrontal cortex (mPFC) for self-referential processes [21,22,23]; the lateral and medial parietal cortex for orienting attentional resources to internal representations, contributing to the re-experience of AMs [24,25], and visual-processing areas including the occipital cortex, cuneus, and precuneus to evoke vivid sensory details and mental imagery [26,27]. 

As regards the emotional component of AMs, recollection of emotionally relevant personal events involves frontotemporal regions and limbic areas such as the amygdala [28,29,30], the hippocampus [31], and the inferior frontal gyrus [32]. Concerning the emotional dimensions of AMs, the intensity of the emotion (arousal) affects the degree to which a personal life event is relived during retrieval and memory vividness [33]. In addition to arousal, the valence of an emotional event can influence how likely and how accurately AMs are remembered [34,35]. Accordingly, neuroimaging showed that brain activity during recall of AMs is modulated not only by arousal but also by the valence of emotion [36,37]. 

Neuroscientific research on AMs with emotional content has developed various methods of mood induction, with autobiographical recall being the most effective compared to other approaches [38]. In autobiographical recall tasks, participants are usually submitted to a pre-scan interview in which they have to select and write down personal events of their lives. Afterward, interviews are reviewed by the experimenter and then presented during the scan session using generic or specific cues to elicit the retrieval (i.e., written instruction, emotionally related words or images, human faces expressing emotions) [38]. Then, participants are guided by the cues to relieve and re-experience their emotional events as vividly and intensively as possible [39].

In 2002, a meta-analysis of neuroimaging studies investigating emotions was conducted [40] including 16 (out of 55) studies using autobiographical recall methods to induce retrieval of personal emotional events (e.g., fear, sadness, happiness, anger, and disgust). Results showed that the anterior cingulate cortex (ACC) and the insula are the brain regions most frequently activated during the recall of AMs, regardless of their valence. Of note, some of these studies highlighted differential brain activations during the recall of AMs corresponding to positive valence. For example, Lane and colleagues [41] showed greater activation of the ventromedial PFC while recalling AMs for happy events compared to AMs for sad events. Accordingly, subsequent studies showed that frontal brain regions (e.g., medial PFC) were more active during retrieval of positive AMs, whereas posterior regions (e.g., right temporal lobe) were more active during retrieval of negative events [36,37]. 

The interest in understanding the processing of positive emotions such as happiness and joy and their neural correlates has increased [42,43,44,45,46,47]. A meta-analysis of imaging studies on happiness was conducted to identify the neural correlates of three happiness domains: pleasure, engagement, and meaning [47]. A wide range of tasks was used to examine these three domains of happiness across the 64 studies included in the meta-analysis identifying 33 brain regions [47]. A further step would be the identification of the brain areas that are specifically involved in the reliving of positive AMs in healthy individuals. This is crucial for understanding their possible alterations in mood disorders: for example, depressed individuals exhibit impaired memory for positive material [48] and recall less vividly positive AMs [49]. In this direction, a work by Suardi and colleagues [50] reviewed 15 neuroimaging studies (7 fMRI, 8 PET) investigating AMs of happy events in healthy individuals, all of them employing autobiographical recall methods. The PFC, ACC, and insula were the most frequently reported areas associated with remembering happy AMs, suggesting that these may be crucial areas implicated in the recall of positive AMs. However, due to the descriptive nature of the review, it was not possible to quantitatively define consistent and significant activation patterns across studies. 

Of note, individual imaging studies, if examined separately, have small sample sizes, thus leading to low statistical power and low reliability [51]. Furthermore, evaluating consistency is important to avoid false-positive rates in the activation locations reported by single studies, which in neuroimaging is relatively high compared to other fields [52]. To overcome these limitations, meta-analysis is a valuable tool for summarizing results and identifying consistently activated brain regions across a set of studies [53]. Therefore, the present study aimed to identify consistent activations across neuroimaging studies of AM for happy life events using Activation Likelihood Estimation (ALE) [54,55], which is one of the most common algorithms used for coordinate-based meta-analysis [53]. To elucidate the neural underpinning of happy AMs, all the included studies were conducted in healthy samples. 

The present meta-analysis differs from those previously described [40,47]. First, we investigated the neural correlates of positive AMs rather than happiness as a broader emotion [47]. Second, the present work includes a significantly larger number of studies on happy AMs than the previous meta-analysis conducted by Phan and colleagues in 2002 [40]. Finally, this is the first meta-analysis to focus on autobiographical recall as a specific type of mood induction procedure, which would help to reduce the differences in brain activation related to the experimental paradigm. 

## 2. Materials and Methods

### 2.1. Information Sources and Search Strategy

This meta-analysis was conducted according to the international guidelines embraced by the Cochrane Collaboration and the “PRISMA” statement to ensure transparent and complete reporting of data selection [56]. A systematic literature search strategy was conducted using the two electronic databases (Web of Science and PubMed/Medline). The search included articles published up to 22 January 2023. Different sets of query terms were adopted: -“autobiographical memory” OR “autobiographical recall”, AND “positive events”, OR “happy events”, AND “fMRI” OR “functional magnetic resonance imaging”.-“autobiographical memory” OR “autobiographical recall”, AND “positive events” OR “happy events”, AND “PET” OR “positron emission tomography”.

Additional sources included published reviews and meta-analyses on neural correlates of emotions and specifically on autobiographical memory for emotional events (e.g., [9,40,46,47]). Most of the studies were selected from a previous review paper on the neural correlates of happiness [50].

### 2.2. Eligibility Criteria

The retrieved papers were analyzed to ascertain that they met the following inclusion criteria: (1) population of healthy adults (2) using autobiographical recall to asses AMs referred to specific events (i.e., episodes lasting between some minutes and one day); (3) including happy AMs and a control condition, such as AM of neutral events or events eliciting emotions with a negative valence (e.g., fear, disgust, sadness); (4) presenting specified neuroimaging acquisition parameters: (a) using whole-brain analysis; (b) reporting the results in Talairach or Montreal Neurological Institute (MNI) coordinates; (5) reporting cerebral activation changes, as assessed by blood-oxygen-level dependent (BOLD) -fMRI or PET. 

We excluded studies that employed non-human participants, clinical populations, as well as those not assessing AM using autobiographical recall techniques or not respecting the neuroimaging parameters. The following publication types were excluded: meta-analysis, systematic reviews, case reports or series, and grey literature. Moreover, articles not written in English were excluded. The authors double-checked the fulfillment of the eligibility criteria. 

### 2.3. Coordinate-Based Meta-Analysis 

ALE was performed using the random effects algorithm of GingerAle (v.3.0.2, http://brainmap.org, accessed date: 22 January 2023) [57,58,59]. Each focus of every experiment is modeled by the ALE as a Gaussian probability distribution:$$p(d)=\frac{1}{\sigma^{3}\sqrt{{\left(2\pi \right)}^{3}}}e^{-\frac{d^{2}}{2\sigma^{2}}}$$
where d indicates the Euclidean distance between the voxels and the considered focus, and σ indicates the spatial uncertainty. The standard deviation is easily obtained through the Full-Width Half-Maximum (FWHM) as follows:$$\sigma =\frac{\mathrm{FWHM}}{\sqrt{8\ln2}}$$

Subsequently, we determined for every experiment a modeled alteration (MA) map as the union of the Gaussian probability distribution of each focus of the experiment. The union of these MA maps provided the final ALE map. The statistical significance of the activation within the ALE map was calculated by cluster-level inference, as suggested by Eickhoff et al. [57,60,61]. Given a particular cluster forming threshold, a null distribution of cluster sizes was obtained by simulating a long series of experiments using the same characteristic of the real data and then by calculating an ALE map. The obtained score histogram was used to assign threshold p values.

### 2.4. Automated Regional Behavioral Analysis

The cluster obtained from the previous ALE meta-analysis was submitted to an automated regional behavioral analysis [62]. Behavioral analysis software was developed and tested as a plug-in application for the Multi-image Analysis GUI (Mango, v. 4.1) image processing system (Lancaster, Martinez; https://mangoviewer.com/api/edu/uthscsa/ric/mango/package-summary.html, accessed date: 22 January 2023). A primary goal of the behavioral analysis is to determine specific behaviors for each region under investigation. The analysis is performed in several steps. For each location in the cluster image obtained from the ALE analysis, a table of behavior domains and sub-domains within each row of the list of coordinates is created. Then, for each location in a behavior coordinate list, a “one” is added and then an image of activation foci by location is created. 

To test for the significance of behaviors, we used the null hypothesis that the observed probability of activation foci was not different from expected, i.e., that po = pe. To determine variance for effect size, we modeled the two possible outcomes of activations (inside or outside of the ROI) using the binomial distribution. In this study, po and pe served as binomial “success” probabilities (probability of activations falling within the ROI) and the number of trials was the whole-brain activation tally (Nb) for a sub-domain. For the binomial distribution, the variance of “p” is calculated as p(1 − p)/N. An effect-size z-score for each behavioral sub-domain was calculated. 

## 3. Results

### 3.1. Study Selection

A total of 365 articles were retrieved from the literature search and other sources. After removing duplicates (n = 100), we conducted title and abstract screening for the remaining 287 studies. This resulted in the exclusion of 243 studies because they were not related to emotional AMs or because they did not use neuroimaging (i.e., fMRI or PET). The remaining 44 studies were screened at a full-text level. Of these studies, 27 were excluded for reasons including (1) no contrasts for happy AM; (2) no coordinates; (3) region of interest (ROI) analysis; (4) only deactivation foci for happy AM; (5) no healthy control sample; (6) review/meta-analysis. A total of 17 studies (12 fMRI, 5 PET) were finally included in the meta-analysis (see Figure 1).

### 3.2. Characteristics of the Studies 

The 17 studies included were published between 1995 and 2019. For each study, the following characteristics are specified in Table 1: the sample size; the neuroimaging technique; the recall induction technique adopted to elicit AMs; the remoteness of the events to remember; which experimental conditions are contrasted to measure brain activation (e.g., happiness vs. neutral); and the number of activation foci (See Table 1). 

The total sample comprised 340 healthy participants ranging in age from 18 to 74 years. Although the majority of studies recruited participants of both genders, six studies included only female participants [63,64,65,66,67,68,69] and one study included only male participants [70], resulting in an overall higher number of females (n = 241) than males (n = 99) in the final sample.

In all the studies, participants were required to re-imagine or mentally recall AMs during the MRI scan. The technique adopted to elicit autobiographical recall differed across studies. In most of the studies, participants are required to recall and relive personally experienced emotional events selected prior to the experimental session. The autobiographical events were cued by written keywords or short sentences in nine studies (see Table 1). In four studies [65,71,72,73], events to be recalled were elicited by listening to pre-recorded audio scripts of autobiographical events. In two studies, the events were cued by pictures depicting emotional facial expressions [63,64], whereas in one study [74], emotional pictures from the International Affective Pictures System (IAPS [75]) served as cues. All the studies adopted a measure to assess the effectiveness of the recall procedure by asking participants to evaluate phenomenological features of AMs (e.g., vividness, intensity, sensory details), although these measures varied across studies. Although neural differences have been suggested during the search and the elaboration phases of the memory process [76], none of the studies included a distinction between these two stages of AM retrieval. 

Eight out of seventeen studies [30,37,65,66,67,72,73,77] considered the remoteness of the event to recall, by asking the participants to evoke events within defined time periods (e.g., the last 5 years). By contrast, in the rest of the studies, participants were free to evoke AMs, without any temporal delimitation.

In eight studies, cerebral activations referred to the contrast between a happiness condition and a neutral condition consisting of recalling autobiographical events without emotional content [64,65,68,71,72,73,78,79], with two studies additionally including the contrast between happiness and irritability [72,73]. In four studies, the control conditions were AMs with negative valence [37,66,67,77]; in one study, AMs related to sadness [63]; and in two studies, AMs related to disgust [74]. Of note, two studies [30,69] reported activations for a happiness condition contrasted with a resting condition, and one study [70] contrasted a happiness condition with a control task (count backward from 100 by subtracting 4). A total of 282 activation foci were reported for all the studies. 

**Table 1 healthcare-12-00711-t001:** Characteristics of the studies and number of activation foci.

Year of Publication	First Author and Reference	Neuroimaging	Original Coordinates	Sample	Recall Induction Technique	Remoteness	Contrasted Conditions	Activation Foci
1995	George [63]	PET	Talairach	n = 11 (F, mean age: 33.3, SD: 12.3)	REC/REL Two events for condition cued with pictures of emotional faces	not defined	happiness > sadness	5
1996	George [64]	PET	Talairach	n = 20 (10 F, mean age: 34.5, SD: 12.1; 10 M, mean age: 35.5, SD: 8.8)	REC/REL Two events for condition cued with pictures of emotional faces	not defined	happiness > neutral	8
1997	Lane [65]	PET	Talairach	n = 12 (F, mean age: 23.3, SD: 3.2)	LIST.SCRIPT Three events for condition	last 6 months	happiness > neutral	4
2000	Damasio [78]	PET	Talairach	n = 41 (21 F, 20 M divided into four cohorts, age: from 23 to 42)	REC/REL One event for condition	not defined	happiness > neutral	20
2003	Markowitsch [36]	fMRI	MNI	n = 13 (7 F, 6 M, mean age: 30, from 19 to 43)	REC/REL 18 events for condition cued by keywords	before 12 years old; from 12 to 18 years; from 18 until now	happiness > rest	10
2003	Piefke [37]	fMRI	Talairach	n = 20 (10 F, 10 M, mean age: 26, SD: 3)	REC/REL 10 events for condition, cued by written sentences	childhood (up to 10 years); recent past (last 5 years)	happiness > negative	4
2007	Marci [71]	PET	MNI	n = 10 (5 F, 5 M, mean age: 33.9, SD: 11.9)	LIST.SCRIPT 2 events for condition	not defined	happiness > neutral	4
2008	Cerqueira [72]	fMRI	Talairach	n = 11 (5 F, 6 M, mean age: 32.4, SD: 7.2)	LIST.SCRIPT 3 events for condition	last 12 months	happiness > neutral happiness > irritability	10
2010	Cerqueira [73]	fMRI	Talairach	n = 11 (5 F, 6 M, mean age: 32.4, SD: 7.2)	LIST.SCRIPT 3 events for condition	last 6 months	happiness > neutral happiness > irritability	5
2011	Sitaram [74]	fMRI	MNI	n = 12 (mean age: 25 years, range: 22–26)	REC/REL 1 event for condition cued by emotional pictures	not defined	happiness > disgust	112
2011	Zotev [70]	fMRI	Talairach	n = 14 (M, mean age: 27.5, SD: 11.1)	REC/REL 3 happy events cued by the word “happy”; counting task as control condition	not defined	happiness > control	22
2014	Speer [79]	fMRI	Talairach	n = 19 (10 F, 9 M, mean age: 26.1, SD: 7.78)	REC/REL 21 episodes for condition cued by keywords	not defined	happiness > neutral	27
2014	Gong [66]	fMRI	MNI	n = 12 (F, mean age: 66.3, from 60 to 70)	REC/REL 10 events for condition cued by written sentences	before 12 years old; last 5 years (except the last month)	happiness > negative	3
2014	Ge [67]	fMRI	MNI	n = 27 (13 younger F, age from 18 to 22; 14 older F, age from 60 to 74)	REC/REL 5 events for condition cued by written sentences	last 5 years	happiness > negative	3
2017	Lempert [69]	fMRI	MNI	n = 35 (F; mean age: 20.86, SD: 2.9)	REC/REL 10 events for condition cued by written sentences	not defined	happiness > rest	12
2018	Xu [77]	fMRI	MNI	n = 25 (17 F, 8 M, mean age: 21.36, SD: 3.34)	REC/REL 9 events for condition cued by written sentences	before 18 years old	happiness > baseline happiness > negative	21
2019	Schie [68]	fMRI	MNI	n = 47 (F, mean age: 29.36; SD: 9.61)	REC/REL 4 events for condition cued by written sentences	not defined	happiness > neutral	12

Notes: SD = standard deviations; n = sample size; F = females; M = males; REC/REL = recalling and reliving past emotional experiences; LIST.SCRIPT = listening autobiographical scripts; MNI = Montreal Neurological Institute.

### 3.3. Clusters of Neural Activity Changes

The brain regions identified in the meta-analysis are presented in Table 2. ALE maps were computed using GingerALE 3.0.2, at an FWE-corrected threshold of *p* < 0.05, with a minimum cluster size of K >150 mm^3^, and visualized using Mango (Figure 2). Eight activation clusters were found. One cluster included the left medial frontal gyrus and the left anterior cingulate (BA 10, 32). Other clusters were identified, including the left posterior cingulate (BA 23, 31), the left superior frontal gyrus (BA 9), and the anterior cingulate (BA 32). Another cluster was found in the left hypothalamus and the left medial globus pallidus. One cluster included the left parahippocampus (BA 35, 36) and parahippocampal structures, and another cluster was found in the left hippocampus and the lateral globus pallidus. Finally, the right thalamus, including the subthalamic nucleus, constituted another cluster of activation. 

### 3.4. Characterization of the Clusters

To characterize in terms of behavior of the various clusters found with the ALE methods, we used a plug-in of the Mango software (v.4.1) [62] that automatically associates the different clusters with the different behavioral domains using the BrainMap database. BrainMap categorizes functional imaging experiments using five major behavioral domains (action, cognition, emotion, interoception, and perception) with 51 sub-domains. Each experiment in BrainMap is assigned one or more behavioral classifications along with the set of coordinates for reported activations, and these data provide the basic structure for forming behavioral probability distributions as 3D images. Region of interest (ROI) analysis is applied to these spatial probability images to assess behaviors. 

The behavioral analysis (BD) was performed with a minimum threshold of activation of 38 foci (labeled as “row counts” in the output of the behavioral analysis plug-in in the Multi-image Analysis GUI). Several BD were identified within the domains of cognition (explicit memory, semantic language, social cognition) and emotion (happiness, fear) (See Figure 3). 

## 4. Discussion

The novel aim of this coordinate-based ALE meta-analysis was to quantitatively analyze the results of neuroimaging studies investigating cerebral activation changes during autobiographical recall of happy life events. This is the first attempt to consider autobiographical recall paradigms assessing happy AMs in a single analysis, to give a more objective perspective of the cerebral network involved when recalling AMs for happy events.

This review included 17 studies (12 fMRI, 5 PET), with an overall number of subjects of 340 for a total of 282 activation foci identified, which is considered a sufficient number to proceed with ALE analysis (as previously stated by Laird in her “Users’ Manual for BrainMap GingerALE 2.0”). 

The outcome of the ALE was characterized by brain regions consistently activated when recalling AMs for happy events. The activation clusters encompassed the frontal gyrus and the cingulate cortex, the basal ganglia, the parahippocampal structures, the hypothalamus, and the thalamus. 

Concerning frontal regions, the prefrontal cortex (PFC) is known to be part of the brain network involved in autobiographical recall processes [18]. Specifically, the medial PFC network, including the anterior and posterior midline regions, has been linked to self-referential processes implicated when recalling personal life events [22,23,80]. Here, we found a consistent activation of the superior and medial frontal gyrus during recall of happy AMs. Previous studies showed that the activation of brain areas within the medial PFC is more frequently associated with happy AMs, whereas activation of the lateral PFC is more frequently associated with sad AMs [30,63,64,81]. In the present meta-analysis, consistent activation of the medial (but not the lateral PFC) seems to support this lateral/medial differentiation in the processing of sad/happy AMs. 

Additional frontal areas were represented by activation in the left anterior cingulate cortex (ACC) and the poster cingulate cortex (PCC). The cingulate cortex activity has been associated with happiness [45,82], although it seems likely that the cingulate cortex plays a key role in both positive and negative emotions [83]. In addition, the cingulate cortex is known to be involved in AMs [10,84,85] and the PCC has particularly strong reciprocal connections with medial temporal lobe memory structures such as the entorhinal and the parahippocampal cortices [86]. Furthermore, the PCC and the medial PFC together with other memory-related areas are part of the “brain default network” [87,88] which is crucial for self-representation and self-consciousness implicated in the recall of AMs. The ACC, closely interconnected to the medial PFC, is known to regulate both cognitive and emotional processing [89,90,91], and consistent activation of the ACC has been found during recall of emotional AMs [40]. 

Concerning the basal ganglia, activation foci were found in the lateral and medial globus pallidus and the subthalamic nucleus. The role of these structures in the reward circuit has been recognized in non-human primates [92,93,94] and more recently in humans [95]. Therefore, the activation of these reward-related areas could be explained by the hedonic features of happy AMs, reflecting hedonic happiness [42,43,47]. 

Finally, the activation found in the parahippocampal regions is consistent with the role of the medial temporal lobe and the hippocampus in AMs retrieval [18], whereas the activation of the hypothalamus and the thalamus may be related to the recall and reliving of sensory and bodily signals associated with happy personal memories [72,73]. 

The majority of clusters were identified in the left hemisphere, aligning with the consistent left-lateralized activation tendency reported in previous neuroimaging literature on autobiographical memory [20,84,96]. This inclination is likely attributed to the verbal modality of autobiographical recall tasks, often involving written keywords or short sentences, thereby implicating the left hemisphere.

Functional characterization of these activations was given by the automated regional behavioral analysis (BD) [62], suggesting that a high portion of the activation foci (n = 157) corresponds to the behavioral domain of emotion, with happiness being the most representative one. Other BD included cognitive domains, mainly represented by episodic memory as expected. A second cognitive domain shown is semantic language, possibly indicating an increase in semantic processing during recall of happy AMs. This may reflect the activation of the PFC and its role in the processing of semantic and conceptual information [35,97,98].

Taken together, the results of the present meta-analysis enable the identification of brain areas consistently activated during the recall of happy AMs. Frontal regions, such as the frontal gyrus and the cingulate cortex, might account for the cognitive processes involved in the subjective appraisal of recalled happy AMs, whereas the subthalamic nucleus and the globus pallidus may be responsible for the pleasure reactions associated with recalling happy AMs. Some of these areas, such as the basal ganglia, the PFC, and the cingulate cortex, align with those previously identified for happiness as a broad emotion [47]. In contrast, other areas, like the amygdala and the insula, did not consistently activate during the recall of happy AMs. This is in line with evidence indicating that the amygdala and the insula are more likely to exhibit increased activation during negative affect than positive affect, despite their responsiveness to both positive and negative affect more than neutral affect [99]. Additionally, early findings from a meta-analysis indicate consistent activation of the amygdala for AMs of fear and the insula for AMs of disgust [40]. 

The present work should be interpreted in light of some limitations. Firstly, only a small number of studies (9 out of 17) included the contrast between positive and negative events, and some studies focused on different types of negative emotional AM (e.g., sadness, irritability). Hence, it was not possible to run a separate analysis for the contrast of positive vs. negative AMs, because the results could be strongly driven by only a few experiments [60,100]. However, a direct comparison between negative and positive AMs would help to investigate and discuss possible overlap and discrepancies in brain activity during the recall of positive vs. negative AMs. Thus, this should be considered as an important limitation and results should be interpreted with caution. 

Secondly, a factor potentially contributing to differential brain activations is related to the remoteness of the evoked AM. So far, some studies have shown greater activation of the medial temporal lobe during retrieval of recent compared to remote episodic memories [37,101,102], although other studies did not confirm this differential activation associated with remoteness [29,31,103,104,105,106,107]. In the current meta-analysis, remoteness was not included as a parameter, given that only eight studies adopted specific temporal information of the AMs (see Table 1), also showing discrepancies in the remoteness of the evoked AMs (e.g., recent vs. remote). Specifically, some studies focused on recent memories from 6–12 months to 5 years, whereas others included remote AMs from adulthood (e.g., from 18 years) or adolescence/childhood (e.g., before 12 years). 

Another limitation is that only activation foci of happy AMs were analyzed, while deactivations were not considered in this study. The decision to exclude deactivations was influenced by the prevalence of articles primarily featuring activations, leaving an inadequate number of articles for a comprehensive meta-analysis of deactivations.

Finally, the demographic variables of the current sample such as the higher proportion of females should be considered when interpreting the results. Previous studies have demonstrated gender differences in AM, which are also reflected in different neural activations during AM recall [108,109,110]. Age is another crucial variable in the context of AM, as healthy aging tends to reduce the episodic richness of AM retrieval [111]. Thus, the wide age range (from 18 to 75) across studies in the meta-analysis may potentially mask age-related differences in the neural correlates of happy AMs that are worth investigating in specific age groups (for example, younger and older people).

Despite these limitations, the results obtained offer new insights into the neural correlates of positive AMs, identifying for the first time consistent brain activations across the existing studies using autobiographical recall methods with happy life events. Identifying the neural correlates of positive AMs in healthy individuals is the first step to extending the basic understanding of dysfunctional processes in affective disorders [112,113]. Individuals with depression have shown impairment in autobiographical memory retrieval, particularly in relation to positive material [48,49,114]. Future studies could investigate functional alterations during recall of positive AMs as possible biomarkers of individuals at risk of developing mood disorders. Furthermore, changes induced by pharmacological or psychological treatments have been described in depression [115]. Similarly, changes in the processing of positive AMs following pharmacological or psychological therapies for depression could be investigated to identify biomarkers of clinical response to treatment. 

## 5. Conclusions

The present coordinate-based ALE meta-analysis identified a set of brain regions that are consistently activated during the recall of happy AMs. This included frontal regions (i.e., prefrontal gyrus and cingulate cortex) exerting cognitive control, basal ganglia structures (i.e., globus pallidus and subthalamic nucleus) related to reward, as well as the memory-related hippocampus and thalamic structures for sensory integration during the recall of AMs. Such a highly distributed brain network encompasses areas that mainly involve the emotion domain and cognitive domains of memory and semantic processing. 

## Figures and Tables

**Figure 1 healthcare-12-00711-f001:**
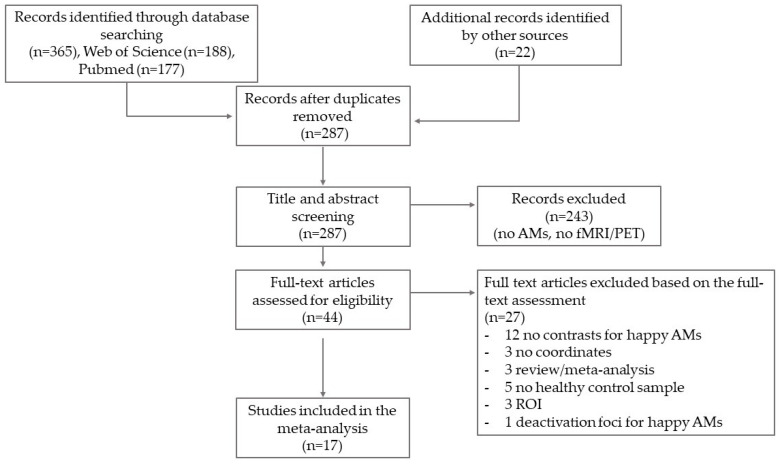
Flow diagram.

**Figure 2 healthcare-12-00711-f002:**
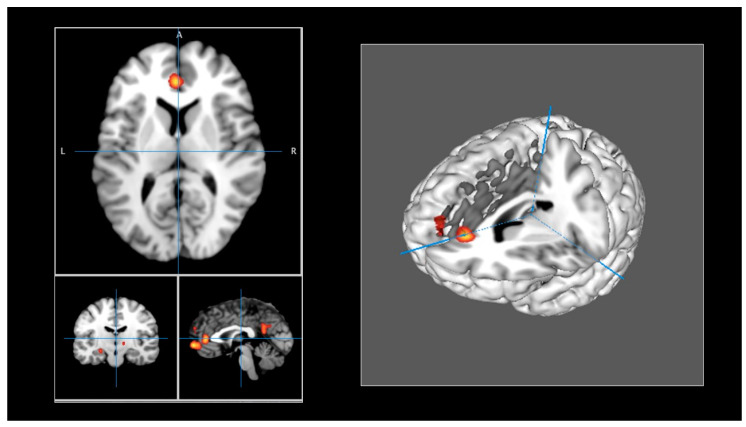
(**Left panel**): ALE maps were computed using GingerALE 3.0.2, at an FWE-corrected threshold of *p* < 0.05, with a minimum cluster size of K > 150 mm^3^, and visualized using Mango. (**Right panel**): Activations were projected onto a 3D rendering model of the brain.

**Figure 3 healthcare-12-00711-f003:**
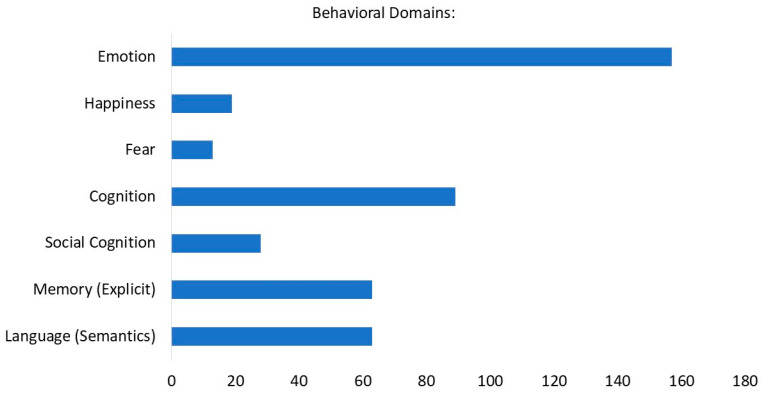
Functional characterization by behavioral domain. The blue bars denote the number of foci for the particular behavioral domain within the selected ROI.

**Table 2 healthcare-12-00711-t002:** Areas of functional changes in brain activity associated with autobiographical recall of happy events.

Cluster	Extrema Value	Side	x	y	z	Label	BA
1	0.0217316690	Left	0	58	−2	Medial frontal gyrus	10
	0.0100130019	Left	−8	40	7	Anterior cingulate	32
	0.0117010213	Left	−2	49	2	Medial frontal gyrus	10
2	0.0196832641	Left	−4	−52	20	Posterior cingulate	23
	0.0100133905	Left	−5	−58	15	Posterior cingulate	23
	0.0145051397	Left	−5	−53	25	Posterior cingulate	31
3	0.0146129345	Left	−8	−6	−6	Hypothalamus	
	0.0100113600	Left	−13	−3	−3	Medial globus pallidus	
	0.0133150452	Left	−10	−5	−6	Hypothalamus	
4	0.0119572980	Left	−2	56	24	Superior frontal gyrus	9
	0.0100121833	Left	−8	56	24	Superior frontal gyrus	9
	0.0116624471	Left	−4	−57	23	Superior frontal gyrus	9
5	0.0149053987	Left	−30	−28	16	Parahippocampus	36
	0.0100117572	Left	−26	−25	−18	Parahippocampus	35
	0.0142512666	Left	−30	−26	−16	Parahippocampus	26
6	0.0134308878	Left	−24	−14	−10	Hippocampus	
	0.0100120860	Left	−21	−15	−9	Lateral globus pallidus	
	0.0134308878	Left	−24	−14	−10	Hippocampus	
7	0.0129249868	Left	−14	20	47	Anterior cingulate	32
	0.0100236097	Left	−14	38	2	Anterior cingulate	32
	0.0123331807	Left	−14	41	0	Anterior cingulate	32
8	0.0127304997	Right	12	−14	0	Thalamus	
	0.0100203901	Right	12	−11	−1	Subthalamic nucleus	
	0.0117313978	Right	12	−13	1	Thalamus	

Notes: *p*-values of the clusters *p* < 0.05.

## Data Availability

The data presented in this study are available on request from the corresponding author.

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
