# Peer review of "The Functional Neuroimaging of Autobiographical Memory for Happy Events: A Coordinate-Based Meta-Analysis"

_healthcare, 2024, doi:10.3390/healthcare12070711_

Round 1

Reviewer 1 Report

Comments and Suggestions for Authors

The manuscript presents the results of a meta-analysis examining the regions activated due to recalling happy events. The analysis included 17 studies with 339 participants (both gender) and had both fMRI and PET studies. The analysis found 8 clusters with areas activated in frontal lobe, thalamus, parahippocampal gyrus, posterior cingulate cortex, hippocampus, and thalamus,a nd globus pallidus.

Comments:

Please include the number of ration of male/female subjects included in meta-analysis. What is the age range of the subjects in studies?

In table 2, I assume that p-values are included (please state so in legend).

In table 3, it is not clear why happiness is so low. Given that criteria where that ‘happiness’ was the search parameter why is it so low – seems similar as to fear.

Have there been previous meta-analysis of happiness?

Have there been other meta-analyses of fear or negative emotion, and how do the results in the current study compare to other meta-analysis of emotion?

Reviewer 2 Report

Comments and Suggestions for Authors

Comments to the Author:

The study by Testa et al. entitled “The Functional Neuroimaging of Autobiographical Memory for Happy Events: A Coordinate-based Meta-analysis” provides a great overview of the neuroimaging studies of happy autobiographical memories. The manuscript is well-written, the aim of the study is clear and the results are in line with the literature on the field. However, some aspects of the manuscript need to be improved.

·       The statistics used and the p values considered in the analysis are never specified. The authors highlighted in line 233 that ‘FWE-corrected threshold, with a minimum 233 cluster size of K >150 mm3’ was used. However, the most important information is missing, thus the p-value used.

·       It is not clear how from 287 studies, 44 studies are remaining. In this case, it would be helpful to clarify why 243 studies were excluded both in the text and in the PRISMA flowchart

·       Why did the authors not consider the deactivation foci? I think it might be interesting to look also at the ‘negative’ results.

·       Did the authors check that all peaks of a study had been thresholded with the same significance level? (e.g., discarding results only obtained with SVC)?

·       It would be useful to include in Table 1 information about the coordinates (TAL or MNI), the acquisition parameters (e.g. magnet strength, sequence duration) used in each study and the neuroimaging software used for the analysis.

·       Are the x,y,z coordinates reported in Table 2 referring to the extrema values coordinates? If so, which is the referring space (TAL or MNI)?

·       How the authors convert from TAL to MNI (or vice versa) should be added to the methods. It is mandatory to have every coordinate in the same space before starting with the analysis. It might also be helpful for readers to know if the authors accounted for any algorithmic differences in the transformation between MNI and Talairach spaces. Be aware that before Lancaster et al.’s (2007) paper on the icbm2tal transform, software applied the Brett transform in their conversion of MNI to Talaraich space. Hence, for studies conducted before the years 2007/2008, authors should take note of using the Brett transform instead of icbm2tal. Gingerale provides options that allow for the conversion based on the Brett transform and some tips can be found here: http://www.brainmap.org/icbm2tal/. Authors are also recommended to identify the algorithm employed by studies published between 2007-2009 and apply the appropriate conversion space in Gingerale.

·       Was the recollection during the MRI only imaginative or did the participant also have to vocally recall the positive experience? In the studies that used cues such as faces, were the subjects asked when they came out of the MRI if they were able to vividly recall a positive memory? In my opinion, the authors should add this methodological information to the text.

·       It is not clear how the authors defined the ROI for the behavioral domain analysis. Did the authors use the clusters derived from the ALE maps? If so, why there is only one figure? I would expect a BD analysis for each cluster.

·       In my opinion, it would be helpful to better discuss how this metanalysis is different from previous studies on the field already published (see ref. 40 and 49)

·       It might be interesting to discuss why there is a lateralization in the results found. Most of the clusters resulting from the analysis are on the left hemisphere, so a digression on why the left hemisphere is more involved in the processing of happy autobiographical memories is needed.

Minors:

·       Line 295: The posterior cingulate cortex acronym is wrong (ACC instead of PCC).

·       Check for typos.

Author Response

Please see the arrachment

Round 2

Reviewer 2 Report

Comments and Suggestions for Authors

The authors correctly answer my questions.